# High-Resolution Computed Tomography and Lung Ultrasound in Patients with Systemic Sclerosis: Which One to Choose?

**DOI:** 10.3390/diagnostics11122293

**Published:** 2021-12-07

**Authors:** Barbara Ruaro, Elisa Baratella, Paola Confalonieri, Marco Confalonieri, Fabio Giuseppe Vassallo, Barbara Wade, Pietro Geri, Riccardo Pozzan, Gaetano Caforio, Cristina Marrocchio, Maria Assunta Cova, Francesco Salton

**Affiliations:** 1Department of Pulmonology, Cattinara Hospital, University of Trieste, 34149 Trieste, Italy; paola.confalonieri.24@gmail.com (P.C.); marco.confalonieri@asugi.sanita.fvg.it (M.C.); pietrogeri@gmail.com (P.G.); riccardo.pozzan@outlook.it (R.P.); gaetano.caforio382@gmail.com (G.C.); francesco.salton@gmail.com (F.S.); 2Department of Radiology, Cattinara Hospital, University of Trieste, 34149 Trieste, Italy; elisa.baratella@gmail.com (E.B.); cristinamarrocchio@gmail.com (C.M.); m.cova@fmc.units.it (M.A.C.); 3Department of Pulmonology, Azienda Sanitaria Universitaria Giuliano Isontina (ASUGI), 34149 Trieste, Italy; fabio.vassallo@asugi.sanita.fvg.it; 4AOU City of Health and Science of Turin, Department of Science of Public Health and Pediatrics, University of Torino, 10124 Torino, Italy; barbarawade@hotmail.com

**Keywords:** systemic sclerosis-associated interstitial lung disease (SSc-ILD), interstitial lung disease (ILD), systemic sclerosis (SSc), diagnostic imaging, high-resolution computed tomography (HRCT), lung ultrasound (LUS)

## Abstract

Imaging plays a pivotal role in systemic sclerosis for both diagnosis management of pulmonary complications, and high-resolution computed tomography (HRCT) is the most sensitive technique for the evaluation of systemic sclerosis-associated interstitial lung disease (SSc-ILD). Indeed, several studies have demonstrated that HRCT helps radiologists and clinicians to make a correct diagnosis on the basis of recognised typical patterns for SSc-ILD. Most SSc patients affected by ILD have a non-specific interstitial pneumonia pattern (NISP) on HRCT scan, whilst a minority of cases fulfil the criteria for usual interstitial pneumonia (UIP). Moreover, several recent studies have demonstrated that lung ultrasound (LUS) is an emergent tool in SSc diagnosis and follow-up, although its role is still to be confirmed. Therefore, this article aims at evaluating the role of LUS in SSc screening, aimed at limiting the use of CT to selected cases.

## 1. Introduction

Systemic sclerosis (SSc) is an autoimmune disease that affects multiple organ systems, including the lungs [1,2,3,4,5,6,7,8,9,10,11,12,13,14,15]. Interstitial lung disease (ILD) is a common manifestation in systemic sclerosis (SSc) and, despite continuous advances in treatment, remains the major cause of death in SSc patients [1,2,3,4]. It is of utmost importance to detect ILD in its very early stages so as to be able to choose the correct treatment regimen [4,5,6,7,8,9,10,11,12,13]. Imaging plays a central role in the management of systemic sclerosis-associated interstitial lung disease (SSc-ILD), and chest high-resolution computed tomography (HRCT) is currently considered the ‘gold standard’ for establishing an ILD diagnosis [14,15,16,17,18]. In early-stage SSc, the most frequent HRCT pattern is nonspecific interstitial pneumonia (NSIP), with predominant ground-glass opacity (GGO) and reticular abnormality. Less frequently, a usual interstitial pneumonia (UIP) pattern is observed, characterised by honeycombing and traction bronchiectasis [14,15,16,17,18,19]. Over time, the radiographic progression evidences the replacement of GGO with honeycombing/traction bronchiectasis and/or bronchiolectasis. The predominant SSc-ILD pattern is NSIP, based on HRTC evaluations and/or biopsy [14,15,16,17,18,19,20,21]. However, HRCT is unsuitable for frequent screening of the lung parenchymal modifications due to high costs and the risks involved in ionizing radiation (particularly in young women with an early SSc diagnosis) [14,15,16,17,18]. Several authors have recently proposed lung ultrasound (LUS) as an imaging technique for the assessment of SSc-ILD even in its early stages, as LUS has a very high negative predictive value [11,22,23,24,25,26,27,28,29]. Moreover, scoring systems have been proposed to quantify the severity of SSc-ILD, and recent research reports the predictive value of LUS [11,22,23,24,25,26,27,28,29]. These observations are opening up new possibilities for the application of LUS in clinical practice, and several preliminary studies have demonstrated reasonable correlations between LUS and HRCT findings in the detection of ILD in established SSc patients [11,22,23,24,25,26,27,28,29]. However, at the time of writing, there is no consensus as to the role LUS plays in the diagnosis and/or prognosis of SSc-ILD. Therefore, this study aims at providing an overview as to the role HRTC and LUS play in SSc-ILD assessment. It also reports the current evidence that supports the use of LUS in daily clinical activity.

## 2. Systemic Sclerosis Associated Interstitial Lung Disease

Systemic sclerosis (SSc) is a connective tissue disease characterised by immune activation, vasculopathy and fibrosis [1,2,3,4,5,30,31,32]. The fibrosis and vasculopathy processes are strictly involved in the development of SSc-ILD [1,2,3,4,5,30,31,32]. The damage of SSc is initiated by microvascular injury, inducing inflammation, an autoimmune response, fibroblast activation and differentiation [13,33]. Activated myofibroblasts perform a series of functions, culminating in an excessive deposition of the extracellular matrix and the development of fibrosis, also at the lung level [33,34,35,36,37].

The European Scleroderma Trials and Research group (EUSTAR) studied 3656 SSc patients, and plain chest radiography evidenced the presence of ILD in about 50% of the patients with the diffuse cutaneous SSc (dcSSc) and in one-third of those with limited cutaneous SSc (lcSSc), particularly in the presence of anti-DNA topoisomerase I antibodies. SSc-specific anti-U3 RNP and anti-Th/To are also associated with SSc-ILD [37,38,39,40,41,42,43].

Although the clinical course of SSc-ILD varies, making diagnosis challenging, there is a high prevalence of lung involvement in patients with an early SSc. Therefore, it is fundamental to make a correct, timely diagnosis to prevent disease progression, as it has a strong impact on the prognosis [2,3]. Indeed, despite recent advances in treatment, the survival of SSc patients is 74.9% at 5 and 62.5% at 10 years from diagnosis, and the presence of ILD has a 2.9-fold higher mortality risk [1,2,3,4]. Therefore, ILD is included in the American College of Rheumatology (ACR)/European League Against Rheumatism Collaborative Initiative (EULAR) joint classification criteria so as to be able to identify SSc in individuals without skin thickening of the fingers [41].

## 3. High-Resolution Computed Tomography in SSc-ILD

As it is able to detect early changes in the lungs and subclinical lung involvement, HRCT is currently the gold standard imaging technique for the evaluation of ILD, including diagnosis and prognosis [13,14,15,16].

### 3.1. High-Resolution Computed Tomography in SSc-ILD: Qualitative Evaluation

The HRCT examination must be performed according to a standard protocol and be correctly defined when reporting the lung alterations observed, using the glossary of terms for thoracic imaging compiled by the Fleischner Society [44,45,46,47]. Nonspecific interstitial pneumonia (NSIP), with a greater proportion of ground-glass opacification (GGO) and a lower proportion of reticulation, is the most frequent HRCT early-stage pattern (Figure 1 and Figure 2).

Less frequently, a usual interstitial pneumonia (UIP) pattern, characterised by honeycombing and traction bronchiectasis, may also be present (Figure 3).

These observations are mainly based on information obtained from HRCT studies. Lung biopsy is not generally necessary as evaluation of the HRCT pattern allows for a good prediction of the underlying histology. Therefore, lung biopsies are reserved for a minority of cases, such as when the HRCT pattern is atypical, or if there is doubt as to the diagnosis, or in the presence of a complication such as cancer. Desai et al. reported on 225 SSc patients and observed that the HRCT findings in SSc closely resembled those of NSIP as they were characterised by a predominance of GGO and fine reticular opacities, although a pattern akin to UIP was occasionally seen. HRCT has also been used to study the longitudinal behaviour of lung disease in SSc [17]. Launay et al. reported on 90 patients with SSc with the predominantly minimal interstitial disease over a 5-year period [18]. They observed that about 50% of patients with GGO abnormalities at initial CT progressed to coarser fibrosis and honeycombing, whereas the remainder remained stable. Conversely, another study reported that only 15% of patients with a normal initial HRCT showed evidence of progression at 5 years [18]. When there are associated reticular abnormalities, which are a common finding in most SSc cases, short-term disease regression has been observed in only a minority of patients, and in the longer term, GGO usually progressed to overt fibrotic change [48,49,50,51,52,53,54]. It is rare to see a reversal of HRCT modifications, and GGO is commonly associated with irreversible disease [48,49,50,51,52,53,54]. Indeed, as an improvement in HRCT findings has been observed in only 5% of patients with GGO and nonfibrotic interstitial opacities, it is reasonable to presume that GGO may represent fibrosis in many SSc cases [48,49,50,51,52,53,54]. Over time, apart from inflammation, fibrosis is the main histologic finding, even GGO, which represents fine reticulation and is rarely reversed but rather replaced later by overt fibrotic findings, such as reticulation or honeycombing/traction bronchiectasis and/or bronchiolectasis [48,49,50,51,52,53,54]. However, HRTC does have some drawbacks, including the resolution limitations of CT, e.g., anatomical structures are disclosed by variations in density measured in spatial units ‘voxels’), X-ray exposure of patients and a lack of longitudinal studies on CT changes in SSc patients with and without treatment [48,49,50,51,52,53,54].

### 3.2. High-Resolution Computed Tomography in SSc-ILD: Quantitative Evaluation

Several recent studies on HRCT have demonstrated that this technique is also able to make a quantification of ILD.

Goh et al. and Warrick et al. proposed various semiquantitative scoring systems to visually quantify the extent of ILD. These can provide prognostic information, e.g., limited or extensive disease and follow-up data in SSc patients with ILD [55,56,57]. The semiquantitative scoring systems can visually quantify the extent of ILD, with a categorical cut-off of 20% to distinguish limited and extensive parenchymal involvement with prognostic implications. More recently, the use of quantitative imaging and artificial intelligence have allowed for objective quantification of ILD through the use of dedicated software, which calculates different parameters of lung density [55]. Several studies have demonstrated the correlation between quantitative and semiquantitative ILD assessment. Yabuuchi et al. [58] observed a significant correlation between the quantitative assessment based on the PFT results and semiquantitative disease extent scores. These results were confirmed by Ninaber et al., Marten et al. and Koyama et al. [59,60,61].

Salaffi et al. [62,63] used the Warrick-score to evaluate the semiquantitative disease extent and reported an excellent correlation between quantitative and semiquantitative disease extent scores. Ariani et al. [64] reported that quantitative lung assessment (QLA) can differentiate mortality risk categories in patients with SSc. They also observed that QLA is useful in the treatment follow-up of SSc patients and the evaluation of pulmonary healing in SSc patients who had autologous stem cell transplantation [58,65,66]. Recently, Bocchino et al. reported that QLA values are significantly correlated with both PFTs and immune parameters (i.e., soluble cytokine receptors) [67].

Very recently, Sambataro et al. have correlated the Quantitative Computed Tomography (QCT) score with disease activity (DA) in SSc patients [68]. The results of this study suggest that QCT can identify SSc patients with a high DA score. This may well open up a scenario of new applications as an operator-independent contribution in DA scores with a potential role for application in clinical practice [69].

## 4. Lung Ultrasound

HRCT is not suitable for frequent screening of the lung parenchymal modifications due to the risks involved in the use of ionizing radiation and high costs. The utility of a lung ultrasound (LUS) in the diagnosis of ILD in very early SSc has also been described and, more recently, its potential for the detection of SSc-ILD in asymptomatic preclinical stages [70,71]. Recent research has focused on the predictive value of LUS [72,73], which is promising for the application of LUS as a screening method for SSc-ILD in clinical practice. Although these are strong arguments in favour of the application of LUS in SSc, to date, there is no unanimous consensus as to the role LUS plays in the diagnosis and/or prognosis of SSc-ILD.

### Lung Ultrasound in SSc-ILD: Qualitative Evaluation

The main LUS signs for ILD-SSc assessment are B-lines and pleural line alterations (Figure 2 and Figure 4). The B-line, generally not present in the lungs of healthy subjects, is defined as a hyperechoic narrow base reverberation artefact, which extends like a laser beam up to the edge of the screen [74,75]. B-lines are excellent markers of pulmonary interstitial involvement, whatever the etiology. However, it must not be forgotten that B-line artifacts have also been reported in patients with cardiogenic pulmonary oedema, interstitial pneumonia (e.g., due to viral infection), acute respiratory distress syndrome (ARDS) and diffuse alveolar haemorrhage [76,77]. It should also be noted that B-line artifacts in patients diagnosed with ILD occur both in actively developing lesions (such as the finding of ground-glass in HRCT) and already existing ones (e.g., honeycombing) [78]. Convex transducers are considered to be the best for the identification of B-lines, especially for the purpose of quantification/semi-quantification [74,75,76,77,78].

When B-lines can be identified singularly, they can be enumerated one by one. However, when they are confluent, such as in the advanced stages of ILD, it is useful to assess the percentage of hyperechoic “white” signal generated by the B-lines below the pleural line and then divide it by 10 [74,75]. Some studies have focused on the assessment of the severity of pulmonary fibrosis based on the number of B-line artifacts [79,80]. Indeed, the counting of B-line artifacts allowed the researchers to make a mathematical correlation with the HRCT results and the application of the Warick score [81,82]. It was reported that the more B-line artifacts were found in each patient, the more severe the pulmonary involvement was. This, in turn, was often associated with a worse gas exchange at pulmonary function tests [83].

Pleural line alterations are quite a common finding in SSc-ILD, especially in the more advanced stages of the disease, whereas only a few B-lines, with no clear irregularities of the pleural line, may be observed in initial lung involvement. These alterations are better visualized by a linear or convex probe with a focus at the level of the pleural line and, as SSc-ILD usually first develops on the posterior lung bases, the LUS examination should focus on this region [74,75,76,77,78].

Recent publications emphasize the need to pay attention to any abnormalities within the pleural line that constitute the source of vertical artifacts [84,85]. These abnormalities can be detected with a linear transducer that has an additional diagnostic value essential in differential diagnosis [86]. Abnormalities within the pleural line include irregular, coarse, fragmented, blurred and/or thickened pleural lines [29,87].

Referring to the data from the 2019 meta-analysis, which analyzed 11 studies involving 487 connective tissue diseases (CTD) patients. The total sensitivity and specificity of the LUS were 0.859 (95% confidence interval (CI) 0.812–0.898) and 0.839 (95% CI 0.782–0.886), respectively. However, most of the studies present the incidence and quantitative or semi-quantitative determination of presence and number of B-lines. The presentation of the ultrasound model of interstitial lung disease is still not precise [88,89,90].

B-lines have a sensitivity of between 59% and 100% and negative predictive value between 51.7% and 100%, while pleural line alterations have a sensitivity between 74% and 85%. Furthermore, B-lines have been reported to have high specificity and a positive predictive value of 59% to 99% and of 90.6% to 95.1%, respectively and pleural lines abnormalities have almost perfect specificity (99% to 100%) [53,54,55,60,70]. Furthermore, LUS is a fast technique (the examination takes, on average, 5–7 min) [88,89,90].

## 5. The Correlation between HRTC and LUS

Several studies have reported a correlation between the frequency of pathological findings in HRCT and LUS [91,92,93]. Indeed, the analysis revealed strong correlations between the observation of a reticular pattern in HRCT and the presence of the irregular, coarse and fragmented pleural lines, as well as single and multiple B-line artifacts visualized at LUS. Interlobular septal thickening was most strongly correlated with the white lung. Bronchiectasis, modified by inflammation, was strongly correlated with an irregular, coarse and fragmented pleural line. Honeycombing was strongly correlated with the presence of consolidations of <5 mm, B-line artifacts forming the white lung and blurred pleural line.

A few studies have reported the correlation between B-lines and vascular damage (capillaroscopic pattern and number of digital ulcers) [12,69,70,71,72,73,74]. Literature data also describe a similar correlation between pleural line alterations (a “US” thickness of >3 mm or irregularities) and HRCT, probably allowing better discrimination than B-lines in ILD detection [94,95,96,97,98]. A negative correlation has been demonstrated between B-lines and pulmonary function tests (PFT) diffusing capacity of carbon monoxide (DLCO) and forced vital capacity) [91,94,99,100,101]. The potential prognostic utility of LUS was explored by Gasparini et al. [91], who showed that basal B-lines predict DLCO change after 12 months of follow-up and by Gargani et al. [92], who demonstrated that a higher number of B-lines is associated with worsening or the development of pulmonary involvement.

## 6. Conclusions

Interstitial lung disease is one of the most common complications in systemic sclerosis and one of the main causes of morbidity and mortality. High-resolution computed tomography is the mainstay for the diagnosis of ILD as it allows for its quantification. Given the exposure to ionizing radiation that the procedure entails, other methods of ILD evaluation are under study, and one of these is lung ultrasound. To date, there is evidence in favour of the use of LUS in the screening of ILD, even in the early phases of the disease and subclinical lung involvement. LUS is fast (the examination takes on average 5–7 min) and has high sensitivity and specificity. Although further prospective studies on a larger population are necessary, the current evidence indicates that thoracic ultrasound can be used during therapeutic follow-up to complete clinical evaluation as it is practical, affordable and without side effects.

The next steps for wider applicability of LUS in the assessment of SSc-ILD are technique standardization (definition of the findings to be used, protocols for image acquisition and the quantification of findings). Moreover, other ‘clinical’ needs should be met, such as its validation in the early stages of the disease, evaluation of the optimal timing for diagnosis and follow-up and the determination of the minimal significant detectable changes.

## Figures and Tables

**Figure 1 diagnostics-11-02293-f001:**
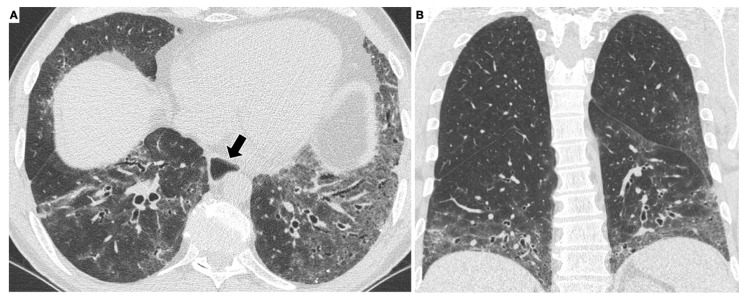
HRCT scan of a patient with systemic sclerosis. The axial scan (**A**) shows diffuse, bilateral, symmetrical ground-glass opacities in the lung parenchyma. The alterations have a basal predominance, as better appreciated on the coronal reconstruction (**B**). Note the dilated esophagus (arrow in **A**), an important accessory finding in this disease.

**Figure 2 diagnostics-11-02293-f002:**
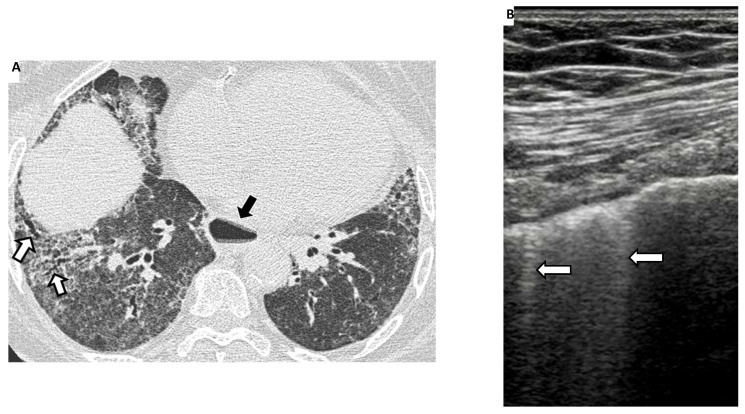
To the left (**A**), an HRCT scan of a patient with systemic sclerosis showing extensive ground-glass opacities and reticulations with traction bronchiectasis (white arrows) in the basal regions of both lungs. The findings are compatible with a fibrotic NSIP pattern. Note also, in this case, a dilated esophagus (black arrow). To the right (**B**), ultrasonographic scans (4–13 MHz broadband linear transducer) with the presence of 2 ultrasound B-line (white arrows) in the same patients.

**Figure 3 diagnostics-11-02293-f003:**
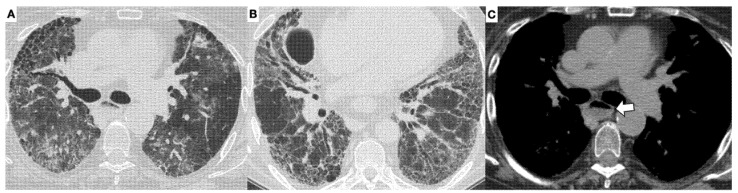
UIP pattern in a patient with systemic sclerosis. On the axial HRCT images at different levels (**A**,**B**), there are bilateral reticular opacities with inter- and intralobular septal thickening, more prevalent in the peripheral regions of the lungs, traction bronchiectasis, ground-glass opacities, and honeycombing. On the axial image in the mediastinal window (**C**), the dilated esophagus, with an air-fluid level within the lumen, can be well appreciated (arrow).

**Figure 4 diagnostics-11-02293-f004:**
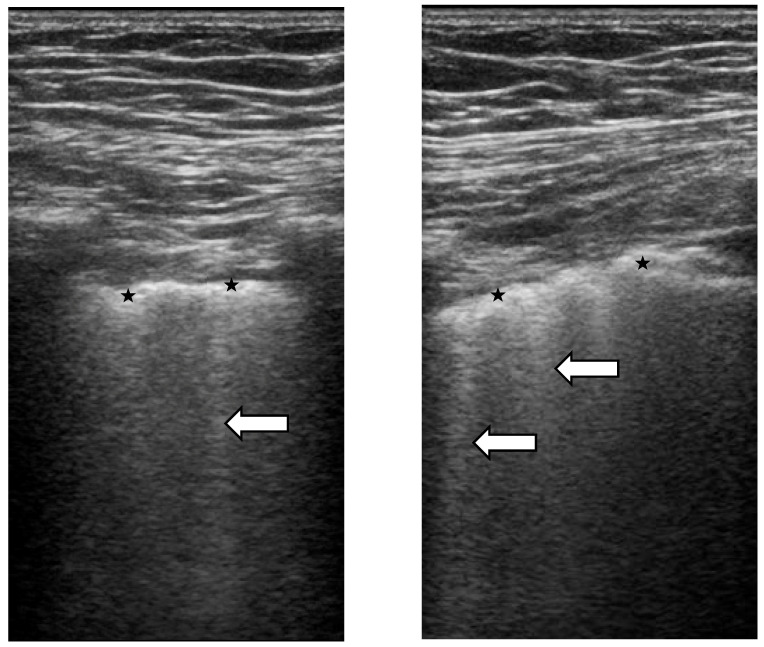
Ultrasonographic scans (4–13 MHz broadband linear transducer). Ultrasound B-lines (or comet-tails) are defined as hyperechogenic artefact consistent with thickened subpleural interlobular septa (white arrows). They originate from the pleural line and are roughly perpendicular to it. They have a narrow base and form a ray that spreads away from the pleural line towards the bottom of the screen and move synchronously with the lung respiration. The pathological pleural line shows irregularities (★).

## Data Availability

Not applicable.

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
