# Peer review of "High-Resolution Computed Tomography and Lung Ultrasound in Patients with Systemic Sclerosis: Which One to Choose?"

_diagnostics, 2021, doi:10.3390/diagnostics11122293_

Round 1
Reviewer 1 Report
The article by RUARO and colleagues is a review of research on the use of HRCT and LUS in the diagnosis of interstitial lung lesions in the course of systemic sclerosis. The authors present the strengths and weaknesses of both imaging techniques in an orderly fashion. They also present a potential horizon for further research. The work contains 104 references, in the vast majority of cases the authors cite current works published within the last decade.
Ultimately, I find the work interesting and valuable, and congratulations to the authors for their efforts to review the work on the topic in question.
Minor comments:
- Line 192 “spreads away from the transducer towards the bottom of the screen” – not from the transducer, but from the pleural line. If the B-line spreads from the above the pleural line, it indicates intradermal emphysema. Pleas correct. Also, “hyperechogenic artifact “ instead of “reflection”. B-line are not reflections.
- As the authors describe in the text changes in the LUS concerning the pleural line, I am asking for the addition of figures showing them, preferably made with a linear probe.
- If it possible for authors to show the HRCT scan and some LUS scans from one single patients with systemic sclerosis?
Author Response
We would like to thank the reviewer for the helpful comments and the Editor for the opportunity to reply to the issues raised. The responses to the reviewer’s comments are as follows:
- Line 192 “spreads away from the transducer towards the bottom of the screen” – not from the transducer, but from the pleural line. If the B-line spreads from the above the pleural line, it indicates intradermal emphysema. Please correct. Also, “hyperechogenic artifact “ instead of “reflection”. B-line are not reflections.
R: In agreement with reviewer’s observation, we have corrected the two errors in the manuscript.
- As the authors describe in the text changes in the LUS concerning the pleural line, I am asking for the addition of figures showing them, preferably made with a linear probe.
R: In agreement with reviewer’s observation, we have added Figure 4 in the manuscript.
- If it possible for authors to show the HRCT scan and some LUS scans from one single patients with systemic sclerosis?
R: In agreement with reviewer’s observation, we have added the HRTC and LUS scans from a single patients with systemic sclerosis in Figure 2.
Reviewer 2 Report
In this paper, the authors write a review of the usefulness of HRCT and LUS in the diagnosis of SSc-ILD. Although LUS is not a frequently performed test in actual practice, this review suggests its usefulness in future practice.
The sensitivity and specificity of LUS in the diagnosis of SSc-ILD was found to be high. HRCT is considered to be a more objective indicator because echoes vary widely among operators. However, LUS is considered to be a useful test in the sense that it is minimally invasive and there is no risk of exposure.
When examining a patient with scleroderma for the first time, it is advisable to take HRCT at least once because of the possibility of complications with ILD, but the question of whether to take it in follow-up many times is troubling. Echoes should be followed by echocardiography to see if there are complications of pulmonary hypertension, and regular follow-up is considered necessary. As a step in this process, LUS could be used to see if ILD is present. If HRCT is taken at the first visit and there is no ILD, then LUS can be performed as a follow-up, which may be one way to utilize LUS in actual practice.
There are not many reviews on LUS, and as a suggestion of the above possibilities, this review will be useful for many clinicians.
Author Response
We would like to thank the reviewer for the comments